# Genetic and Environmental Structure of Altruism Characterized by Recipients in Relation to Personality

**DOI:** 10.3390/medicina57060593

**Published:** 2021-06-08

**Authors:** Juko Ando, Tetsuya Kawamoto

**Affiliations:** 1Faculty of Letters, Keio University, Tokyo 108-8345, Japan; 2Faculty of Letters, Kokushikan University, Tokyo 154-8515, Japan; ktetsuya@kokushikan.ac.jp

**Keywords:** altruism, kin selection, direct and indirect reciprocity, twin, genetics

## Abstract

*Background and Objectives:* Altruism is a form of prosocial behavior with the goal of increasing the fitness of another individual as a recipient while reducing the fitness of the actor. Although there are many studies on its heterogeneity, only a few behavioral genetic studies have been conducted to examine different recipient types: family members favored by kin selection, the dynamic network of friends and acquaintances as direct reciprocity, and strangers as indirect reciprocity. *Materials and Methods:* This study investigated the genetic and environmental structure of altruism with reference to recipient types measured by the self-report altruism scale distinguished by the recipient (the SRAS-DR) and examine the relationship to personality dimensions measured by the NEO-FFI with a sample of 461 adult Japanese twin pairs. *Results:* The present study shows that there is a single common factor of altruism: additive genetic effects explain 51% of altruism without a shared environmental contribution. The genetic contribution of this single common factor is explained by the genetic factors of neuroticism (N), extraversion (E), openness to experience (O), and conscientiousness (C), as well as a common genetic factor specific to altruism. Only altruism toward strangers is affected by shared environmental factors. *Conclusions:* Different types of altruistic personality are constructed by specific combinational profiles of general personality traits such as the Big Five as well as a genetic factor specific to altruism in each specific way.

## 1. Introduction

In an evolutionary context, human altruism refers to the motivation to increase the fitness of another person as a recipient while reducing one’s own fitness (Nowak, 2006 [1]; de Waal, 2008 [2]). Psychologically, it is the drive toward cooperation between kin and group members in both a narrow (e.g., friends and acquaintances) and broad (e.g., strangers in one’s own community) perspective. Altruism is typically conceptualized as a component of the broader construct of “prosocial behavior” and sometimes both terms are used as synonyms.

Altruistic behaviors can be observed in numerous forms in everyday life from offering a comforting word, donating an enormous amount of money, or even sacrificing one’s life for a stranger. Several studies have investigated the underlying structure of altruism and prosocial behavior to explain these behavioral diversities (e.g., Padilla-Walker and Carlo, 2014 [3]). Various conceptual subtleties are observed in relation to altruism, namely empathy at an emotional level (Eisenberg, 2006 [4]); sympathy, compassion, benevolence (Schwarz, 2010 [5]) at the cognitive level; and helping, sharing, comforting at the behavioral level (Knafo et al., 2018 [6]). Most studies indicate a single-factor structure, but some studies have reported uncorrelated relationships among factors (Bryant and Crockenberg, 2018 [7]).

According to the research findings on economic games, including the dictator game (DG), the ultimatum game (UG), and the public goods game (PGG), although the level of altruistic behaviors differs based on the characteristics of social groups, altruistic behaviors are common in humans (Henrich et al., 2005 [8]). There are a few who pay nothing to recipients/responders in any culture. The reason why altruistic behavior is adaptive, even though it does not seem to immediately benefit the actors’ survival and reproduction, is because of the probability that individuals who acted in an altruistic way or shared the same genes with those altruistic actors stand to benefit in the long term, resulting in an increase in their inclusive fitness (Trivers, 1985 [9]).

However, this does not mean that all individuals are uniformly altruistic. Cultural and individual differences have been reported in the DG and the UG (Henrich et al., 2005 [8]), and there are large individual differences in the way altruism is shown. The reason for such a wide range of individual differences in altruism, while it has universal adaptive significance, is unclear. It could possibly be because of individual differences in social learning situations or genetic variations. Behavioral genetics would help provide an answer to this question.

A behavioral genetic approach is effective in describing the underlying structure of phenotypic altruism because it can divide phenotypic variability into genetic and environmental structures. Behavioral genetic studies on altruism and prosocial behavior show mixed results. The majority of studies follow the so-called three laws of behavioral genetics: (i) (additive) genetic (A) and (ii) non-shared environmental contributions (E), which are unique to individuals, are substantial; and (iii) shared (common) environmental effects (C), which make family members resemble each other due to environmental influences, are minimal or even nil. The so-called “AE model”, which explains total phenotypic variance only by the additive genetic and non-shared environmental effects, best fits actual data (e.g., Rushton, Fulker, Neale, Nias, and Eysenck, 1986 [10] for adults; Gregory, Light-Hausermann, Rijsdijk, and Eley, 2009 [11] for adolescence; Knafo-Noam, 2015 [12] for childhood). Although these twin studies have been conducted in some European countries, North America, and Australia, behavioral genetic studies on prosocial behavior in South Korea (Hur, 2007 [13]) and Nigeria (Hur et al., 2017 [14]; Hur et al., 2018 [15]) also reported that the AE model is the best fit.

Twin studies sometimes reported the “ACE model” as the best fit model, in which about 50% of total variance is explained by the additive genetic effects (A), around 20% by shared family environment (C), and 30% by non-shared or unique environments (E) (e.g., Stevenson, 1997 [16] for children and adolescents; Wang and Saudino, 2015 [17] toddlers).

Several studies, however, showed the “CE model”, in which individual differences of altruism and prosocial behavior are explained by the environment, without any genetic contribution (e.g., Kruger, Hicks, and McGue, 2001 [18] for adults; Lemery and Goldsmith, 2002 [19]). The best fitting model depends on the type of behavior. Knafo-Noam, 2018 [6] reported that the CE model fits for helping (C = 22%) but the AE model is the best fit for sharing (A = 24%) and comforting (A = 50%) in childhood.

One reason for these inconsistent results may be the developmental change in genetic effects and environmental structure. Generally, shared environmental effects decline and genetic effects increase with age (Scoufield et al., 2004 [20]).

Another reason for these heterogeneous results can be attributed to the various recipients of altruistic behavior. In many studies, adult recipients are strangers, but friends and siblings are usually recipients of toddlers for children. From an evolutionary perspective, differences in recipients are important because of the genetic relatedness between the actor and recipient. Altruistic behavior toward family members can help the actor’s own genes, which is called “kin selection” (Hamilton, 1964 [21]), whereas non-kin altruism seems to have no direct benefits immediately for the actor’s genes but has evolved into the so-called phenomenon of “reciprocity” (Trivers, 1971 [22]). Reciprocal behavior towards an actor’s friends or acquaintances in a community may have potential benefits, incurring costs in the present if a reverse situation happens in the future in which the recipient who was previously helped by the actor may extend help, which is called “direct reciprocity”. Altruistic behavior towards strangers may also have a potential beneficial effect if reciprocal behavioral “purchases” increase cooperation from others through a good reputation (Nowak and Sigmund, 1998 [23]), which is called “indirect reciprocity”.

Different types of altruism could be affected differently by psychological characteristics, such as personality traits, in relation to their functional significance, by which an individual takes some specific survival strategy to adapt to their social environment.

For example, kin selection is hypothesized to relate to empathy, whereas non-kin reciprocity is related to forgiveness (Ashton et al., 1998 [24]. Ben-Ner and Kramer, 2010 [25] showed that personality was not related to giving money to kin in a dictator game but a significant relation to strangers (collaborators, neutrals, and competitors).

Oda et al., 2013 [26] developed an altruism scale distinguished by three recipient types (refer to Section 2.2) and reported correlational patterns between those three types of altruism and the Big Five dimensions of personality based on a sample of Japanese university students. Traits such as extraversion have been found to substantially predict all three types of altruism. Openness to experiences and agreeableness were also found to relate significantly but only slightly to the three types of altruism. Conscientiousness correlated slightly only with altruism toward family members. It is worthwhile to replicate these findings in an adult sample from the same country, as well as to investigate genetic and environmental structures by the twin method.

A meta-analysis of the relationship between the cognitive process of economic games (UG and DG) and personality affordances (Thielmann et al., 2021 [27]) suggested that altruism toward strangers might be a one-shot interaction between actors and recipients, which is afforded by agreeableness (A), such as in DG. On the contrary, temporal conflict that might occur in repeated interactional situations, such as between friends and acquaintances (indirect reciprocity), is afforded by conscientiousness (C).

This study will investigate more sophisticated pictures of the genetic and environmental structure of altruism with reference to recipient types and examine its relationship to personality dimensions in a sample of the Japanese population.

## 2. Materials and Methods

### 2.1. Participants

The study sample was drawn from the Keio Twin Study (KTS), a more than 20-year longitudinal twin cohort study conducted in Japan, beginning in 1998. The data structure and the method for zygosity diagnosis were reported by Ando et al., 2013 [28] and Ando et al., 2019 [29]. Analysis of data of 461 adult twin pairs, including 258 female monozygotic (MZ) twin pairs, 69 male MZ pairs, 73 female dizygotic (DZ) pairs, 15 male DZ pairs, and 46 opposite sex DZ pairs, were analyzed. The age range was from 22 to 49 years with a mean of 30.85 years and SD of 5.50.

### 2.2. Measurement

#### 2.2.1. Altruism

The self-report altruism scale, which is distinguished by the recipient, the SRAS-DR (Oda et al., 2013 [30]; Oda et al., 2014 [26]), was used to measure the altruistic behaviors enacted in daily life and to classify them according to the recipient (family members for kin selection, friends or acquaintances for direct reciprocity, and strangers for indirect reciprocity). Respondents were asked to rate the frequency, in their life in the past, with which they have engaged in altruistic behavior using five categories (“never”, “just once”, “several times”, “often” to ‘‘very often’’). To construct the SRAS-DR, 21 items were used—seven items for each recipient. The example items for family members are: (i) I have supported one of my family members when they were not feeling well; and (ii) I have helped one of my family members when the person was overburdened. Examples for friends or acquaintances are: (i) I have listened to the troubles and complaints of a friend/acquaintance; and (ii) I have accompanied a friend to a place they wanted to go. Examples for strangers are: (i) I helped a stranger who fell on the road; and (ii) I helped a stranger put his/her luggage on a train or bus rack. Reliability indices (Cronbach’s α) were 0.807 for family members, 0.812 for friends and acquaintance, and 0.819 for others, showing adequate internal consistency.

#### 2.2.2. Personality Dimensions

The Japanese version of the NEO Five-Factor Inventory (NEO-FFI) (Costa and McCrae, 1992 [31]; Yoshimura et al., 2001 [32]) and a shortened version of the NEO-PI-R were employed to measure personality dimensions. The NEO-FFI is a 60-item inventory that provides a reliable measure of the Big Five (BF) dimensions of personality (Neuroticism (N), Extraversion (E), Openness to experiences (O), Agreeableness (A), and Conscientiousness (C)). As the study participants have been participating in the longitudinal cohort project (KTS), they took the NEO-FFI three times. To maximize the number of cases and to obtain stable scores, the averages of all scores that each participant had gotten were employed in the following analysis. Reliability indices (Cronbach’s α) were 0.870 for N, 0.862 for E, 0.635 for O, 0.714 for A, and 0.820 for C, showing adequate internal consistency.

### 2.3. Data Analysis

To calculate the descriptive statistics of altruism and personality score, IBM SPSS statistics version 26.0 was used. To conduct structural equation modeling (SEM) to obtain latent estimates of the additive genetic factors, shared, and non-shared by univariate, multivariate, and Cholesky decomposition, Mplus ver. 7 (Muthén and Muthén, 2012 [33]) was used. We used the following three fit indices to evaluate model fit: the comparative fit index (CFI; Bentler, 1990 [34]); the root mean square error of approximation (RMSEA; Steiger, 1990 [35]); and the standardized root mean square residual (SRMR: Bentler, 1995 [36]). Previous research suggested that values of CFI > 0.95, RMSEA < 0.06, and SRMR < 0.08 are regarded as a good fit (Hu and Bentler, 1999 [37]).

## 3. Results

### 3.1. Phenotypic Correlation

Table 1 shows the phenotypic correlations among the three dimensions of altruism, age, gender (0 = female, 1 = male), and the Big Five dimensions of personality by zygosity. Age was significantly positively correlated with altruism toward family members and strangers for MZ twins and was significantly positively correlated with altruism toward family members for DZ twins. Additionally, age was significantly negatively associated with altruism toward friends/acquaintances for MZ twins. Regarding gender differences in altruism, females significantly scored higher in altruism toward family members and friends/acquaintances than males for both MZ and DZ twins. Moreover, several significant relationships with age and gender were found in the Big Five traits. Therefore, in the following analysis, age and gender effects were eliminated for altruism scores and the Big Five personality scores using the multiple regression analysis technique because these effects might be confounding factors to elevate twin correlations (McGue and Bouchard, 1984 [38]).

### 3.2. Univariate Analysis

The first two columns of Table 2 show the twin intra-class correlations of MZ and DZ. The MZ intra-class correlations were higher than DZ intra-class correlations for all three types of altruism and the Big Five personality dimensions, indicating a substantial genetic contribution.

The results of univariate analysis that estimate relative contributions of the additive-genetic effects (A), shared environment (C), and non-shared environment (E) are summarized in the remaining columns of Table 2 with 95% confidence intervals (CI). The three types of altruism, neuroticism (N), and agreeableness (A) provided a small amount of shared environment, but their 95% CI included zero. There is no C estimated in extraversion (E), openness to experience (O), and conscientiousness (C).

### 3.3. Comparison between Common and Independent Pathway Models

To examine the genetic and environmental structure of altruism, the independent pathway (IP) model and the common pathway model (CP) were compared. Both CP and IP are statistical models that explain multiple intercorrelated phenotypic variables using more simple and parsimonious latent genetic and environmental factor structures. The IP model assumes latent genetic and environmental factors that are independent of each other and load on each variable with a specific weight. The CP model assumes an underlying latent common factor (or factors) affected by genetics and environmental effects that load on each variable in specific ways; it suggests some common psychological functions that affect multiple related behavioral traits and is a nested model of IP.

The likelihood ratio test suggested that the CP model did not significantly harm the fit to the present data compared to the IP model. Additionally, despite a slightly lower CFI value than that of the other two models, the CP model showed a better fit to the data based on AIC, BIC, and RMSEA (Table 3). Therefore, it was selected as the best-fit model (Figure 1). Figure 1 shows the results for the CP model. There was no significant contribution of shared environment to a common factor, altruism toward family members, or altruism toward friends/acquaintances. Only altruism toward strangers had a significant effect on the shared environment.

The finding of the CP model as the best-fit model suggests that there is a single general psychological mechanism underlying altruism toward different types of recipients. This single common factor is affected by genes and non-shared environment but not by shared environments. Specific altruistic factors toward family and friends/acquaintances are also explained only by genetics and non-shared environment. However, altruism toward others is affected by shared environment as well as by genetics and non-shared environments.

### 3.4. Genetic and Environmental Relationships between Personality and Altruism

Genetic and environmental relationships between personality traits and the three types of altruism were examined using the Cholesky decomposition technique (Figure 2). Because the Cholesky decomposition model including additive genetic, shared environmental, and non-shared environmental effects did not converge, and because the univariate genetic analysis showed there were no shared environmental effects on all the variables, we applied the Cholesky decomposition model including additive genetic effects (A) and non-shared environmental (E) effects to the eight variables (three altruistic behaviors and five personality traits). The fit indices of this Cholesky decomposition model with a χ^2^ of 241.833 and 224 degrees of freedom (*p* = 0.197) were quite good (CFI = 0.993, RMSEA = 0.015, and SRMR = 0.055). The parameter estimates are presented in Table 4.

With the exception of A, personality dimensions (N, E, O, and C) explain the overlap of the three types of altruism, which may consist of a single common factor of altruism. More specifically, the genetic factors of altruism toward family were characterized primarily by genetic overlap with C (0.277), as well as with N (−0.144), E (0.197), and O (0.118); there was no genetic overlap with A. Genetic factors of altruism toward friends/acquaintances were characterized by genetic overlap with all the five personality dimensions, with E being the most common (0.371), as well as N (−0.221), A (0.184), O (0.184) and C (0.131). Genetic factors of altruism toward others were characterized by genetic overlap primarily with E (0.340), as well as with N (−0.182), O (0.118), and C (0.179); no genetic overlap was observed with A. The genetic factor of A was only significantly related to altruism toward friends.

The effect of a non-shared environmental structure bridging personality and altruism appears weaker than that of additive genetic structures. No significant relationships were found, except for correlations between E and family, E and friends/acquaintances, and A and family. Non-shared environmental interrelationships between the three types of altruism were greater than that between personality and altruism.

Additionally, we observed novel genetic and non-environmental contributions to altruistic behavior, which cannot be explained by personality-related genetic and non-environmental factors. Specifically, a genetic factor of family significantly loaded on friends/acquaintances and strangers, as well, showing a general affect across all three different types of altruism.

## 4. Discussion

The results of the present study replicate a single-factor structure of multiple facets of altruistic behavior from the perspective of recipient difference. Genetic factors explain nearly 51% of the variance of this single latent factor of altruism, and the rest is explained by a non-shared environment, which is consistent with the literature. The shared environment factor is negligible for this common latent factor of altruism. The genetic and environmental structure of altruism is quite similar to that of prosocial behavior in childhood (Knafo-Noam et al., 2015 [10]).

Behavioral genetic studies usually indicate that non-shared environmental effects are trait-specific (or opportunity-specific). Although the present study has replicated this pattern, there were significant overlaps via non-shared environmental factors among the three types of altruism.

The present study has also replicated E and O, which commonly contribute to all three types of altruism of recipients (Oda et al., 2013 [26]), as well as N and C at the genetic level. In particular, genetic overlap with E appears dominant among these four personality dimensions. The other personality dimensions except for A also contributed toward each type of altruism in a genetically specific way. The genetic factor of A only overlapped with altruism toward friends/acquaintances, which characterized the genetic profile of direct reciprocity in a specific way. As for altruism toward family, no genetic, but significant non-shared environmental overlap was found with A.

The finding that there are relatively weak genetic overlaps of altruism toward different types of recipients with major personality dimensions, such as the Big Five, is theoretically important because it suggests that basic personality traits partly underlie the variation in altruism. The “altruistic personality” has often been debated. The present study also suggests that different types of altruistic personality are constructed by specific combinational profiles of general personality traits such as the Big Five in each specific way. Altruism-specific genetic factors have been discovered, which are independent from those specific to personality traits. The proportions of variances explained by additive genetic effects specific to altruistic behaviors were much larger than those explained by additive genetic effects specific to personality traits. This indicates that altruistic behaviors have different evolutionary roots from basic behavioral tendencies. Animal studies, such as those on birds, have shown that different types of personalities (e.g., neophobic or neophilic) adopt different styles of defense against predator strategies, and not all personality types can be involved in reciprocity in animals (Vrublevska et al., 2014 [39]). Both altruism and personality can ultimately increase the probability of survival and reproduction. However, the environmental tasks that these two traits can solve are different, which may lead to small genetic overlaps between altruism and personality traits.

Another important finding is that only altruism toward strangers is affected by the shared environment. This suggests that it is possible to alter altruism toward others by informing them about strangers at risk or in helpless situations and teaching skills to help those strangers. On the contrary, altruistic attitudes toward family members and friends/acquaintances are difficult to alter by educational treatment because they are a kind of personality and are not constructed based on knowledge and skills.

The finding that the genetic influence of A had a significant genetic relationship only with altruism toward friends/acquaintances and not toward family members and strangers appears strange because A is semantically the core concept of all kinds of altruism, and there is a significant phenotypic correlation between A and the three altruistic variables (Table 1). The result of the Cholesky decomposition indicated that these phenotypic correlations stem from genetic correlations with N, E, O, and C, suggesting that altruistic tendencies toward family members and strangers are not based on A, which is an interpersonal attitude, but are characterized as a different psychological background in regard to human relations.

### Limitation

The limitations of the present study are acknowledged. First, because of the limited and unbalanced sample size, especially with a small number of male participants, gender differences in genetic and environmental structures could not be examined. Although significant mean differences between females and males were observed in the present sample, we could not conduct sex-limitation models. By using sex-limitation models, we can investigate sex differences in expression of genetic and/or environmental factors (Neale, Røysamb, and Jacobson, 2006 [40]). Future research might want to extend the present findings by conducting sex-limitation models using a large sample.

Second, although the study sample was involved in a longitudinal cohort project, change and stability of genetic and environmental structures were not investigated. This was because altruism has been measured once up to now. By measuring it in the following waves, we will be able to investigate this.

Finally, the findings of the present study were exclusively obtained using self-report questionnaire data and not by experimental or real situations. Behavioral data, such as economic games and observational data, will be necessary to confirm these findings.

Despite these limitations, the current study indicates several possibilities for future research. The result of a common pathway model suggested the existence of a general factor of altruism (GFA); therefore, it is an interesting research question to investigate the relationships between GFA and other general psychological factors such as general intelligence (g) and a general factor of personality (GFP) (Musek, 2007 [41]). In particular, because GFP is considered as an evolutionarily relevant characteristic (Rushton, Bons and Hur, 2008 [42]), this relationship might be important. It is also an important question to see sex differences in altruism from an evolutionary point of view because different sexes might have different strategies for altruistic behavior. Investigation of shared environmental influence toward altruism for strangers (indirect reciprocity) is another important question to be addressed because this altruistic factor is directly and indirectly related to socially important values such as benevolence, human rights, equality, and so on.

In conclusion, the present study shows that a single-factor structure among different types of altruism, according to the recipients, is a genetic mixture of personality traits such as extraversion, openness to experiences, and conscientiousness. Altruism toward strangers is the only type that is affected by a shared environment of agreeableness.

## Figures and Tables

**Figure 1 medicina-57-00593-f001:**
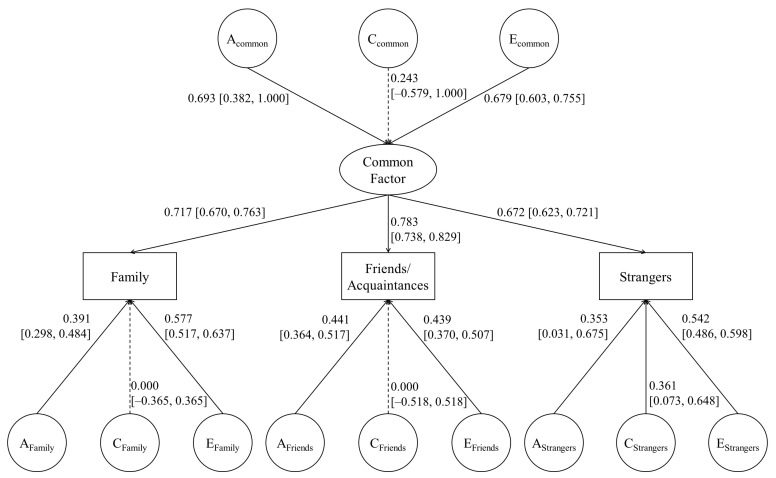
Common pathway model of altruism. A_common_, C_common_ and E_common_ represent additive genetic factor and non-shared environment in the common pathway model. A_Family_, C_Family_, E_Family_, A_Friends_, C_Friends_, E_Friends_, A_Strangers_, C_Strangers_, and E_Strangers_ represent specific A, C, and E factors of altruism for family, friends/acquaintance and others.

**Figure 2 medicina-57-00593-f002:**
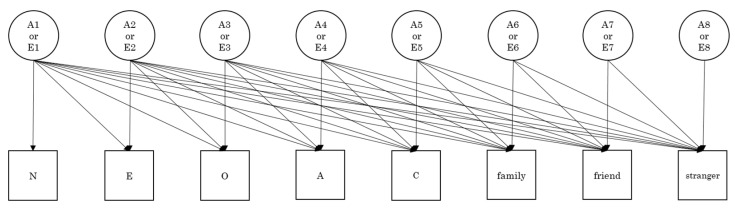
Cholesky decomposition model analyzing genetic and environmental relationships between three types of altruism and big five personality dimensions (N = neuroticism, E = extraversion, O = openness to experience, A = agreeableness, C = conscientiousness). Full (saturated) model was employed for additive genetic (A) and non-shared environment (E) factors (not shown in this figure but the same as A). A reduced model was employed for shared environment factors based upon the results of univariate and common pathway models.

**Table 1 medicina-57-00593-t001:** Phenotypic correlations among all variables by zygosity.

	Age	Gender	Family	Friend	Stranger	N	E	O	A	C
Age		−0.004 [−0.099, 0.092]	0.191 [0.085, 0.293]	0.032 [−0.077, 0.140]	0.045 [−0.063, 0.153]	−0.350 [−0.454, −0.238]	0.104 [−0.019, 0.224]	−0.054 [−0.176, 0.069]	0.036 [−0.087, 0.159]	0.062 [−0.061, 0.183]
Gender	0.088 [0.024, 0.151]		−0.154 [−0.258, −0.046]	−0.129 [−0.234, −0.021]	0.002 [−0.107, 0.110]	−0.237 [−0.350, −0.117]	0.046 [−0.077, 0.168]	0.002 [−0.121, 0.125]	−0.053 [−0.175, 0.070]	0.121 [−0.002, 0.240]
Family	0.136 [0.065, 0.207]	−0.096 [−0.168, −0.024]		0.518 [0.434, 0.593]	0.508 [0.423, 0.585]	−0.037 [−0.175, 0.102]	0.239 [0.103, 0.365]	0.057 [−0.082, 0.194]	0.267 [0.133, 0.391]	0.132 [−0.007, 0.266]
Friend	−0.151 [−0.221, −0.079]	−0.226 [−0.294, −0.156]	0.559 [0.507, 0.607]		0.483 [0.395, 0.562]	−0.007 [−0.145, 0.132]	0.351 [0.223, 0.467]	0.144 [0.005, 0.277]	0.280 [0.146, 0.403]	0.155 [0.017, 0.288]
Stranger	0.097 [0.024, 0.168]	0.048 [−0.025, 0.120]	0.474 [0.416, 0.528]	0.508 [0.453, 0.560]		−0.146 [−0.279, −0.007]	0.318 [0.188, 0.437]	0.259 [0.124, 0.384]	0.169 [0.031, 0.301]	0.248 [0.113, 0.374]
N	−0.125 [−0.201, −0.048]	−0.135 [−0.210, −0.058]	−0.161 [−0.245, −0.076]	−0.180 [−0.263, −0.095]	−0.182 [−0.264, −0.096]		−0.449 [−0.542, −0.345]	0.000 [−0.123, 0.123]	−0.234 [−0.347, −0.114]	−0.400 [−0.498, −0.291]
E	−0.121 [−0.197, −0.044]	−0.008 [−0.085, 0.070]	0.261 [0.179, 0.341]	0.450 [0.378, 0.516]	0.309 [0.229, 0.386]	−0.378 [−0.443, −0.310]		0.104 [−0.019, 0.224]	0.316 [0.201, 0.422]	0.343 [0.229, 0.447]
O	−0.148 [−0.223, −0.071]	0.051 [−0.027, 0.128]	0.069 [−0.018, 0.155]	0.168 [0.082, 0.251]	0.156 [0.070, 0.239]	0.037 [−0.040, 0.114]	0.069 [−0.008, 0.146]		0.045 [−0.079, 0.167]	0.062 [−0.062, 0.183]
A	−0.002 [−0.080, 0.075]	−0.032 [−0.109, 0.046]	0.226 [0.142, 0.307]	0.322 [0.242, 0.398]	0.119 [0.033, 0.204]	−0.300 [−0.368, −0.227]	0.300 [0.228, 0.369]	0.101 [0.023, 0.177]		0.173 [0.051, 0.289]
C	0.089 [0.012, 0.165]	0.045 [−0.033, 0.122]	0.309 [0.228, 0.385]	0.283 [0.201, 0.361]	0.262 [0.179, 0.341]	−0.383 [−0.448, −0.315]	0.338 [0.268, 0.405]	0.064 [−0.014, 0.140]	0.250 [0.176, 0.321]	

Notes. MZ = monozygotic twins, DZ = dizygotic twins, N = neuroticism, E = extraversion, O = openness to experience, A = agreeableness, C = conscientiousness. 95% confidence intervals are shown in parentheses. Phenotypic correlations below the diagonal are for MZ and those above the diagonal are for DZ.

**Table 2 medicina-57-00593-t002:** Twin intra-class correlations and estimated contributions of additive genetic (A), shared environment (C) and non-shared environment (E).

	MZ	DZ	A	C	E
Family	0.449 [0.355, 0.532]	0.148 [−0.042, 0.314]	0.437 [0.349, 0.525]	0.000 [0.000, 0.000]	0.563 [0.475, 0.651]
Friend	0.498 [0.409, 0.575]	0.243 [0.068, 0.395]	0.468 [0.143, 0.792]	0.034 [−0.262, 0.331]	0.498 [0.418, 0.579]
Stranger	0.533 [0.449, 0.606]	0.359 [0.194, 0.496]	0.378 [0.050, 0.706]	0.155 [−0.150, 0.460]	0.467 [0.391, 0.544]
N	0.501 [0.425, 0.570]	0.332 [0.177, 0.467]	0.353 [0.046, 0.660]	0.148 [−0.138, 0.434]	0.499 [0.429, 0.569]
E	0.553 [0.482, 0.616]	0.262 [0.102, 0.405]	0.555 [0.491, 0.619]	0.000 [0.000, 0.000]	0.445 [0.381, 0.509]
O	0.509 [0.434, 0.577]	0.251 [0.088, 0.396]	0.519 [0.451, 0.588]	0.000 [0.000, 0.000]	0.481 [0.412, 0.549]
A	0.380 [0.294, 0.459]	0.273 [0.113, 0.415]	0.206 [−0.168, 0.580]	0.161 [−0.189, 0.512]	0.633 [0.553, 0.712]
C	0.504 [0.429, 0.573]	0.292 [0.133, 0.431]	0.514 [0.444, 0.584]	0.000 [−0.012, 0.012]	0.486 [0.417, 0.555]

Notes. MZ = monozygotic twins, DZ = dizygotic twins, A = additive genetic, C = common (shared) environment, E = non-shared environment, CI = confidential interval, N = neuroticism, E = extraversion, O = openness to experience, A = agreeableness, C = conscientiousness. 95% confidence intervals are shown in parentheses.

**Table 3 medicina-57-00593-t003:** Results of model comparison.

	*χ* ^2^	*df*	*p*	Δχ^2^	Δdf	*p*	AIC	BIC	RMSEA	90% CI	CFI
Cholesky (Full)	54.928	33	0.010				7344.125	7437.318	0.046	[0.023, 0.067]	0.980
Independent pathway (Full)	54.701	33	0.010	–	–	–	7343.898	7437.090	0.046	[0.022, 0.067]	0.980
Common pathway (Full)	60.273	37	0.009	5.572	4	0.233	7341.470	7416.911	0.045	[0.023, 0.065]	0.978

**Table 4 medicina-57-00593-t004:** Standardized estimates and their 95% confidence intervals (CIs) of Cholesky decomposition model on the big five personality dimensions and three recipient types of altruism.

Additive Genetic Effect								
	N	E	O	A	C	Family	Friend	Stranger
N	0.713 [0.666, 0.761]							
E	−0.366 [−0.449, −0.283]	0.652 [0.601, 0.703]						
O	−0.026 [−0.121, 0.069]	−0.026 [−0.123, 0.071]	0.722 [0.674, 0.770]					
A	−0.207 [−0.296, −0.119]	0.156 [0.068, 0.244]	0.005 [−0.082, 0.093]	0.551 [0.484, 0.617]				
C	−0.315 [−0.401, −0.229]	0.155 [0.066, 0.243]	−0.043 [−0.131, 0.045]	0.071 [−0.034, 0.176]	0.620 [0.564, 0.675]			
Family	**−0.144** [−0.245, −0.044]	**0.197** [0.097, 0.297]	**0.118** [0.017, 0.219]	0.073 [−0.049, 0.195]	**0.277** [0.173, 0.380]	**0.538** [0.448, 0.628]		
Friend	**−0.221** [−0.318, −0.123]	**0.371** [0.279, 0.463]	**0.136** [0.039, 0.232]	**0.184** [0.068, 0.301]	**0.131** [0.029, 0.234]	**0.188** [0.063, 0.313]	**0.449** [0.369, 0.529]	
Stranger	**−0.182** [−0.283, −0.081]	**0.340** [0.245, 0.436]	**0.228** [0.130, 0.326]	−0.107 [−0.229, 0.014]	**0.179** [0.074, 0.285]	**0.270** [0.145, 0.395]	0.080 [−0.041, 0.200]	**0.469** [0.393, 0.546]
**Non-Shared Environmental Effect**								
	**N**	**E**	**O**	**A**	**C**	**Family**	**Friend**	**Stranger**
N	0.701 [0.653, 0.749]							
E	−0.192 [−0.252, −0.131]	0.636 [0.590, 0.682]						
O	0.004 [−0.060, 0.068]	0.122 [0.058, 0.187]	0.680 [0.631, 0.729]					
A	−0.201 [−0.271, −0.130]	0.117 [0.047, 0.187]	0.081 [0.011, 0.151]	0.754 [0.706, 0.802]				
C	−0.182 [−0.246, −0.118]	0.119 [0.057, 0.182]	0.053 [−0.010, 0.115]	0.022 [−0.039, 0.083]	0.660 [0.612, 0.708]			
Family	−0.013 [−0.094, 0.067]	**0.128** [0.046, 0.209]	0.022 [−0.061, 0.105]	**0.090** [0.009, 0.171]	−0.004 [−0.083, 0.076]	0.728 [0.671, 0.785]		
Friend	−0.043 [−0.119, 0.032]	**0.100** [0.024, 0.177]	**0.099** [0.021, 0.176]	0.032 [−0.044, 0.109]	−0.011 [−0.086, 0.063]	0.362 [0.294, 0.430]	0.594 [0.545, 0.643]	
Stranger	0.004 [−0.067, 0.075]	0.063 [−0.009, 0.136]	0.048 [−0.026, 0.121]	0.030 [−0.043, 0.102]	0.024 [−0.047, 0.096]	0.233 [0.164, 0.301]	0.240 [0.176, 0.303]	0.580 [0.534, 0.627]

Notes. N = neuroticism, E = extraversion, O = openness to experience, A = agreeableness, C = conscientiousness. Significant correlation coefficients (i.e., 95% CI does not contain zero) between the big five dimensions and three altruism types are bolded.

## Data Availability

The data that support the findings of this study are available on request from the corresponding author, J.A. The data are not publicly available due to their containing information that could compromise the privacy of research participants.

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
