# Peer review of "Genetic and Environmental Structure of Altruism Characterized by Recipients in Relation to Personality"

_medicina, 2021, doi:10.3390/medicina57060593_

Round 1

Reviewer 1 Report

Dear Authors.

Thank you for addressing my comments. I can conclude that you substantially improved the text. However, before endorsing publication of this manuscript, I would like to see your reaction to the following recommendations.

Major comments

(1) Explain what "functional significance" is (line 100), and if necessary, expand the sentence that includes this term.

(2) Provide the values of Cronbach's for each Big Five dimension measured with NEO-FFI. If possible, indicate that a calculation of the facets was not possible. For example, some researchers may need this information when carrying out meta-analyses on personality trait and their facets.

(3) For Table 2, indicate whether the values of the effects were unique or not (i.e., whether included error terms or not).

(4) In paragraph 2.3 briefly explain how the values of CFI, RMSEA, and SRMR were interpreted in terms of good/excellent fit; provide a suitable reference.

(5) Provide at least two ideas for future research.

Minor comments

(6) Correct the numbering of tables.

(7) Ensure that CIs are presented consistently (e.g., compare Table 2 and 3 this one with the correlations, and correct accordingly).

(8) Explain why accounting for sex in the analysis would have been necessary. Note that not everyone knows that standard practice in this kind of study is constraining the effects (means) across men and women and then test model fit.

Kindest regards.

Author Response

Dear Authors.

Thank you for addressing my comments. I can conclude that you substantially improved the text. However, before endorsing publication of this manuscript, I would like to see your reaction to the following recommendations.

Major comments

(1) Explain what "functional significance" is (line 100), and if necessary, expand the sentence that includes this term.

According to the reviewer’s comment, we have added, this term has been expanded to

“their functional significance by which an individual takes some specific survival strategy to adapt to social environment.”

(2) Provide the values of Cronbach's for each Big Five dimension measured with NEO-FFI. If possible, indicate that a calculation of the facets was not possible. For example, some researchers may need this information when carrying out meta-analyses on personality trait and their facets.

According to the reviewer’s comment, we have added the following sentence.

“Reliablity indices (Cronbach’s α) were .803 for N, .712 for E, .624 for O, .711 for A, and .819 for C, showing adequate internal consistency.”

(3) For Table 2, indicate whether the values of the effects were unique or not (i.e., whether included error terms or not).

According to the reviewer’s comment, we have revised Table 2 including 95% CI for twin correlations.

(4) In paragraph 2.3 briefly explain how the values of CFI, RMSEA, and SRMR were interpreted in terms of good/excellent fit; provide a suitable reference.

According to the reviewer’s comment, we have added the following text to the paragraph 2.3:
“We used the following three fit indices to evaluate model fit: the comparative fit index (CFI; Bentler, 1990) [35]; the root mean square error of approximation (RMSEA; Steiger, 1990) [36]; and the standardized root mean-square residual (SRMR: Bentler, 1995) [37]. Previous research suggested that values of CFI > 0.95, RMSEA < 0.06, and SRMR < 0.08 are regarded as good fit (Hu & Bentler, 1999) [38].”

In addition, we have added the following references:

“35. Bentler, P. M. (1990). Comparative fit indexes in structural models. Psychological Bulletin, 107, 238–246.”

“36. Steiger, J. H. (1990). Structural model evaluation and modification: An interval estimation approach. Multivariate Behavioral Research, 25, 173–180.”

“37. Bentler, P. M. (1995). EQS structural equations program manual. Encino, CA: Multivariate Software.”

“38. Hu, L., & Bentler, P. M. (1999). Cutoff criteria for fit indexes in covariance structure analysis: Conventional criteria versus new alternatives. Structural Equation Modeling, 6, 1–55.”

(5) Provide at least two ideas for future research.

According to the reviewer’s comment, we have revised the text in the Limitation from:

“The limitations to the present study are acknowledged. First, because of the limited sample size, especially with a small number of male participants, gender differences could not be examined. Significant mean differences between females and males in the present sample, different genetic and environmental structures have been found.

Second, although the study sample is involved in a longitudinal cohort project, developmental trends such as change and stability of genetic/environmental structure were investigated.

Finally, the findings of the present study were exclusively obtained using questionnaire data and not by experimental or real situations. Behavioral data, such as economic games and observational data, will be necessary to confirm these findings.”

to

“The limitations to the present study are acknowledged. First, because of the limited and unbalanced sample size, especially with a small number of male participants, gender differences in genetic and environmental structures could not be examined. Although significant mean differences between females and males were observed in the present sample, we couldn’t conduct sex-limitation models. By using the sex-limitation models, we can investigate sex differences in expression of genetic and/or environmental factors (Neale, Røysamb, & Jacobson, 2006 [40]). Future research might want to extend the present findings by conducting the sex-limitation models using a large sample.

Second, although the study sample has been involved in a longitudinal cohort project, change and stability of genetic and environmental structures were not investigated. This was because altruism has been measured once up to now. By measuring it in the following waves, we will be able to investigate that.

Finally, the findings of the present study were exclusively obtained using self-report questionnaire data and not by experimental or real situations. Behavioral data, such as economic games and observational data, will be necessary to confirm these findings.

Despite these limitations, the current study indicates several possibilities for future research. The result of a common pathway model suggested existence of a general factor of altruism (GFA), then it is an interesting research question to investigate relationships between GFA and other general psychological factors such as general intelligence (g) and a general factor of personality (GFP)(Musek, 2007)[41]. Especially, because GFP is considered as an evolutionarily relevant characteristics (Rushton, Bons & Hur, 2008)[42], this relationship might be important. It is also an important question to see sex differences in altruism from evolutionary point of view because different sex might take different strategy for altruistic behavior. Investigation of shared environmental influence toward altruism for strangers (indirect reciprocity) is another important question to be addressed because this altruistic factor is directly and indirectly related to socially important values such as benevolence, human rights, equality, and so on.”

In addition, we have added the following references:

“40. Neale, M. C., Røysamb, E., & Jacobson, K. (2006). Multivariate Genetic Analysis of Sex Limitation and G × E Interaction. Twin Research and Human Genetics, 9, 481–489.”

“41. Musek, J. (2007). A general factor of personality: Evidence for the Big One in the five-factor model. Journal of Research in Personality, 41, 1213–1233.”

“42. Rushton, J.P., Bons, T.A., & Hur, Y-M (2008) The genetics and evolution of the general factor of personality. Journal of Research in Personality, 42, 1173–1185.”

Minor comments

(6) Correct the numbering of tables.

This error has been corrected in accordance with the reviewer's comment.

(7) Ensure that CIs are presented consistently (e.g., compare Table 2 and 3 this one with the correlations, and correct accordingly).

According to the reviewer’s comment, we have added the 95% confidence intervals to Tables 1 and 2. In addition, we have revised Table 1 to show the phenotypic correlations by zygosity.

(8) Explain why accounting for sex in the analysis would have been necessary. Note that not everyone knows that standard practice in this kind of study is constraining the effects (means) across men and women and then test model fit.

According to the reviewer’s comment, we have added the following phrase;

 because these effects might be confounding factors to elevate twin correlations (McGue & Bouchard, 1984) [39]

In addition, we have added the following reference:

“39. McGue, M. & Bouchard, T.J. (1984). Adjustment of twin data for the effects of age and sex. Behavior Genetics, 14, 325–343.”

Kindest regards.

Reviewer 2 Report

Dear Authors,

I have no comment left. Well-done!

Sincerely,

Indrikis Krams

Author Response

Thank you.

This manuscript is a resubmission of an earlier submission. The following is a list of the peer review reports and author responses from that submission.

Round 1

Reviewer 1 Report

Dear Author,

I find your research fascinating. In the manuscript, you attempted to contribute to the current knowledge related to predictors of altruism, which is of fundamental importance in the pursuit of understanding human social behavior. With this, I am suggesting general and specific improvements that could address my concerns.

Major comments

(1) Introduction lacks in nuance. I do not see a reasonable justification for how personality traits can fit the ACE framework. Could you elaborate on this? Notably, the Introduction structure could benefit from additional edits that could make the argumentation clearer and nuanced. For example, this fragment "Different types of altruism could be affected differently by psychological characteristics, such as personality traits, concerning their functional significance." (lines 80-81) is vague because "functional significance" was not specified. Besides, what patterns between the constructs could be proposed?

(2) The links between the environmental factors and helping behavior and connections between psychological factors and prosocial behavior have recently been studied in meta-analyzes. For instance, Fisher et al. (2011) showed the impact of situational factors reducing helping behavior; Fromell et al. (2020) studied the effect of different mindsets (impulsive vs. reflective) on altruism; finally, Thielemann et al. (2020) published a paper to indicate the connection between personality and prosocial behavior in a variety of tasks. None of these seminal papers was discussed adequately in your manuscript. Indeed, the main reason to discuss the given papers is not to reflect upon psychological theories of altruism, but rather to report how strongly altruism is related to personality and to what extend can be explained by situational factors (effects aggregated across many studies). This background could be used as a suitable context for an interpretation of your results.

(3) Provide more fit statistics. What was the value of CFI/TLI, RMSEA (plus CIs), chi-squared, and df? Were the values of AIC compared? Was the difference statistically significant?

(4) Discussion does not include any theoretical reflection. Given the findings and the population studied, does this research contribute to any theory or specific model? How?

(5) Limitations are superficial. Note that altruism can be studied through the means of a questionnaire and in a task. This https://cooperationdatabank.org shows the possibilities. I believe that this tool can inspire you to expand your limitations and plan future studies.

Minor comments

(6) I cannot find the value of Cronbach's alpha. What was the reliability level of personality and altruism?

(7) Figure 2 is somehow difficult to read. I suggest labeling the variables A1, A2…, C1,… etc. differently, namely, in a way to be consistent with the other parts of the manuscript. Importantly, does the presentation of the variables in circles and squares imply latent and observed variables, respectively, as in Figure 1? Maybe adding layers or additional labels could improve the readability of Figure 2. 

(8) Table 3 is challenging to read since the major labels on the left-hand side are not well-displayed. Besides, add all values to Table 3 or choose another way for presenting all the findings. For meta-analytic purposes, all effects must be reported.

(9) The content of Table 3 requires a good explanation of how to read the values. Are the values any type of weights? A reader should know how to interpret values on the diagonal, e.g., O-O = 3.770, and other values, e.g., N-E = 2.990 or N-friend 0.112.

(10) Do you have data for personality facets? If yes, then I recommend presenting them in an Appendix to illustrate the connections between traits' facets and altruism, similarly to the findings in Table 1. This can be quite important for meta-analytic purposes.

(11) In your Discussion, you may want to focus on some more practical problems that can be resolved based on the knowledge of altruism. Currently, many countries and communities need to deal with pollution; for instance, plastics' consumption puts a strain on the recycling process and increases environmental damage in countries that cannot afford to recycle. Given the current findings, how a government could encourage citizens to do something for other people, mostly strangers, to protect the environment and prevent mass extinction?

Conclusion

Overall, you focused on an exciting problem, and you studied it with a sophisticated research technique. However, I do not find this manuscript, at least in the current form, to be of sufficient quality to contribute to the given field. Since a few major concerns need to be addressed, I regrettably cannot recommend publication of this manuscript in the "Medicina" journal, suggesting major revision.

Kindest regards.

References

Fischer, P., Krueger, J. I., Greitemeyer, T., Vogrincic, C., Kastenmüller, A., Frey, D., et al. (2011). The bystander-effect: A meta-analytic review on bystander intervention in dangerous and non-dangerous emergencies. Psychological Bulletin, 137(4), 517–537.
Fromell, H., Nosenzo, D., & Owens, T. (2020). Altruism, fast and slow? Evidence from a meta-analysis and a new experiment. Experimental Economics, 23(4), 979–1001.
Thielmann, I., Spadaro, G., & Balliet, D. (2020). Personality and prosocial behavior: A theoretical framework and meta-analysis. Psychological Bulletin, 146(1), 30–90.

Reviewer 2 Report

medicina-1115035

Ando: Genetic and environmental structure of altruism characterized by recipients in relation to personality.

I enjoyed reading this excellent article on relationships between personality types and altruistic behavior. I can see only some minor issues listed below.

Line 41: Delete a double quotation mark from (E”).

Lines 86-94: Vrublevska et al. 2014 Acta Ethol 18: 111-120 showed that not all personality types can be involved in reciprocity in animals. Perhaps, this may help in explaining the author’s point in light of evolution.

Line 95: Remove “Thus,” as redundant.

Line 114: The author mentions “never” and “very often.” What are the rest of the three categories used in this study?

Line 154-157: Although this is explained in lines 154-157, the legend of Table 1 should be self-explanatory, and the meaning of N, E, O, A, C (neuroticism, agreeableness…) must be mention in the legend of Table 1.

Table 2, Table 3, Figure 1 and 2: the same applies to the rest of the tables (for example, Table 2: the first column needs to be explained to the reader).

Line 237: “could be found” or “have been found”?

Line 244: Remove “5” from agreeableness.